# Sodium Silicate Improves Cucumber Seedling Growth and Substrate Nutrients and Reduces Heavy Metal Accumulation in Plants

Wei Tian [1],[†], Zhaoxuan Li [1],[†], Kaixuan Gong [1], Xiaodong Wang [1], Sadiq Shah [2], Xiaozhuo Wang [1] and Xueyan Zhang [1],*

[1] College of Agriculture, Ningxia University, Helanshan Xilu No. 489, Yinchuan 750021, China; tianweinxu@163.com (W.T.)
[2] Department of Food Science and Technology, Garden Campus, Abdul Wali Khan University, Mardan 23200, Pakistan
* Correspondence: zhangxueyan123@sina.com
[†] These authors contributed equally to this work.

**Abstract:** The gasification filter cake (GFC) has great application potential for improving the characteristics of seedling substrates due to its nutrient richness and excellent water retention capacity. However, GFCs leach heavy metals easily and thus pose certain ecological risks. Sodium silicate can enhance plant resistance to heavy metal toxicity by fixing heavy metals. This study investigated the impact of sodium silicate on cucumber plant growth, the chemical characterization of the substrate, and the distribution and transfer of heavy metals. Sodium silicate was added to the seedling substrate mix at mass rates of 0 g/kg$^{-1}$ (GFC0), 2 g/kg$^{-1}$ (GFC2), 4 g/kg$^{-1}$ (GFC4), and 8 g/kg$^{-1}$ (GFC8). The seedling substrate was composed of a commercial matrix, caragana compost, and GFC (m:m 7:7:2). The GFC increased the content of total phosphorus (P), available phosphorus (P), and available potassium (K) in the substrate by 31.58%, 16.58%, and 80.10%, respectively. Conversely, the GFC decreased the plant height by 12.3%. Adding sodium silicate to the GFC increased the chlorophyll content of the plants, fixed heavy metals in the substrate, and promoted nutrient absorption and utilization by the plants. Compared with GFC0 without sodium silicate, adding sodium silicate at a mass rate of 2 g/kg$^{-1}$ (GFC2) reduced the chromium, lead, and cadmium contents by 51.13%, 26.37%, and 90.04%, respectively, which effectively alleviated heavy metal stress and was more conducive to plant growth.

**Keywords:** sodium silicate; substrate nutrients; distribution of heavy metal




## 1. Introduction

Coal gasification technology is essential for the conversion of coal into cleaner fuel and for improving its utilization during power generation [1,2]. However, a large amount of coal gasification slag is produced during the cleaning process. Every year, China produces about 80 million tons of coal gasification slag and hence, the effective use and safe disposal of these industrial residues is essential [3]. Currently, the major disposal method for gasification filter cakes (GFCs), one of the main by-products of coal gasification slag, is directly into landfills [4,5]. The disposal of GFCs containing high organic carbon (C), nitrogen (N), and phosphorus (P) threatens the environment and may lead to groundwater pollution. Thus, alternative disposal means are necessary to protect the environment from the ill effects of GFCs [6,7].

A gasification filter cake (GFC) has a small particle size and large specific surface area and contains silicates of aluminum (Al) and calcium (Ca), which facilitate the stabilization of heavy metals in waste residue. However, under the influence of low external pH values, a GFC can cause different degrees of heavy metal leaching into the substrate [8,9]. The

ability of a GFC to accumulate nutrients and retain a high capacity of water has prompted studies into its secondary utilization in agricultural fields, especially for improving the characteristics of the seedling matrix [10,11]. However, the transformation from soil to plant and the leaching of heavy metals in a GFC should be strictly controlled for its effective secondary utilization.

Many economical and robust adsorbents, such as carbonaceous materials (biocarbon, nanocarbon), microorganisms, and sodium silicate, have been developed [12,13]. Carbonaceous materials and microorganisms can chelate heavy metals in the soil through adsorption, ion exchange, complexation, precipitation, and enzymatic transformations [14–16]. However, due to their low removal efficacy for heavy metals, poor adsorption, high input cost, and non-sustainability, the application of biosorbents in fields with high concentrations of heavy metals is limited [13,17]. Sodium silicate was found to be an effective compound that can reduce the bioavailability of Cd and Cr by increasing the soil pH and can form silicate complexes to reduce the availability of heavy metals in the soil [18]. It can also resist the toxic effects of heavy metals by improving the antioxidant capacity of plants. Moreover, sodium silicates also increase biomass and the intensity of photosynthesis, thus promoting plant growth and development [19,20].

Littleleaf Peashrub (*Caragana microphylla*) is a local domesticated species that is a perennial C3 leguminous shrub distributed extensively in China. It covers an area of approximately 7 million ha in the southern mountainous region of Ningxia [21]. However, to maintain productivity and reduce disease infestation, *C. microphylla* must be cut at approximately 10 cm aboveground and harvested every three to five years. As a result, an average of about 3 million tons of *C. microphylla* stubble shoots are produced each year. Since the branches of *C. microphylla* can increase the organic matter and N content during the composting process, it has the potential to reduce nutrient input and improve nutrient-use efficiency in crop production [22,23]. Furthermore, the combination of *C. microphylla* compost and other soil conditioners can effectively reduce the nutrient deficiency caused by continuous cropping. This combination can improve the physical and chemical properties of the soil and increase the number and activity of microorganisms in the soil during the cultivation of horticultural crops [23].

The use of caragana compost relieves the accumulation of *C. microphylla* residues and reduces the use of chemical fertilizers; this greatly reduces environmental pollution and promotes sustainable agriculture [24]. Therefore, *C. microphylla* compost has potential for use as an effective substitute for the seedling matrix and for promoting seedling growth of horticultural crops. Adding a GFC may also accelerate the composting process, reduce C and N loss, and increase the K and P content in the compost [11].

Cucumber is one of the most widely grown vegetables in the world and has tremendous economic value. Its ability to tolerate heavy metal stress is poor, which can easily lead to yield reduction and heavy metal pollution.

In this study, the contribution of sodium silicate to heavy metal risk was evaluated by adding it to the seedling substrate, along with a GFC and *C. microphylla* compost. The objectives of this study were to quantify the effects of sodium silicate on the following: (1) the growth of cucumber seedlings, (2) the substrate nutrient content, and (3) the accumulation and transfer of heavy metals in plants and substrates. We provide a theoretical foundation for the efficient and safe use of agricultural waste straw and coal waste resources.

## 2. Materials and Methods

### 2.1. Experimental Design

This study was conducted in the No. 2 greenhouse at the Ningxia University Training Base located in Xixia District, Yinchuan City, Ningxia, China, between 10 July and 30 August 2017. During the experimental period, the average temperature of the greenhouse was 28 °C and the average humidity was 60 percent. The seedling substrate was combined with *C. microphylla* straw compost and commercial substrate (Ningxia Zhongqing Agricultural Technology Co., Ltd. (Ningxia, China)) at 1:1 (m/m). GFC was added accord-

ing to the mass ratio of 8:1. Various sodium silicate treatments were designed at mass rates of 0, 2, 4, and 8 g/kg, respectively. Treatments without GFC and sodium silicate served as the control (Table 1). *Cucumis sativus* (cucumber) ('Deer No. 99') seeds were cultivated in 72-hole seeding trays (Ningxia Zhongqing Agricultural Technology Co., Ltd. (Ningxia, China)) (50 × 34 × 12 cm, 1 seed per hole) and sown on 15 July 2017. The seeding tray was randomized with three replicates of 15 plants for a total of 45 plants per treatment. Each treatment was provided with 450 mL of water at 24 h intervals from sowing to the end of the experiment (6-true-leaves stage) The seedlings were grown under natural lights (day:night at 16:8 h).

**Table 1.** Experimental treatments for cucumber seedling substrates (mass ratio).

| Treatment | Commodity Substrate% | Caragana Compost% | Sodium Silicate (g kg$^{-1}$) | Gasified Filter Cake (g kg$^{-1}$) |
|---|---|---|---|---|
| CK | 50.00 | 50.00 | -- | -- |
| GFC$_0$ | 43.75 | 43.75 | -- | 12.5 |
| GFC$_2$ | 43.75 | 43.75 | 2 | 12.5 |
| GFC$_4$ | 43.75 | 43.75 | 4 | 12.5 |
| GFC$_8$ | 43.75 | 43.75 | 8 | 12.5 |

Note: '--' in the table indicates zero addition.

The composite seedling matrix was analyzed regarding several physicochemical indices, including pH (6.81), electrical conductivity (0.57 mS cm$^{-1}$), organic matter (11.23 g kg$^{-1}$), total N (1.03 g/kg), available N (15.03 mg/kg), available P (1.21 mg/kg), and K content (21.05 mg/kg). The compost was made with straw from the leguminous shrub *C. microphylla* and sheep manure at a C:N ratio of 25:1. The *C. microphylla* compost had a bulk density of 0.25 g/cm$^3$, total porosity of 79.2%, organic matter of 475 g/kg, total N, P, and K of 29.0 g/kg, 18.6 g/kg, and 3 g/kg, respectively, a pH of 7.86, and EC of 1.66 mS/cm. The GFC was supplied by the Ningxia Coal Industry Group, and the nutrients and heavy metal contents are shown in Table 2.

**Table 2.** Basic nutrient and heavy metal characteristics of the gasification filter cake.

| Item | Gasified Filter Cake | Heavy Metal Standard in Soil (mg kg$^{-1}$) | | |
|---|---|---|---|---|
| | | pH < 6.5 | 6.5 < pH < 7.5 | pH > 7.5 |
| Organic matter (%) | 34.33 | -- | -- | -- |
| pH | 7.90 | -- | -- | -- |
| Total phosphorus (µg $^{-1}$) | 1669.61 | -- | -- | -- |
| Available phosphorus (µg g$^{-1}$) | 107.46 | -- | -- | -- |
| K$^+$ (µg g$^{-1}$) | 5972.56 | -- | -- | -- |
| Total nitrogen (µg g$^{-1}$) | 703.00 | -- | -- | -- |
| Pb (µg g$^{-1}$) | 45.75 | 250 | 300 | 350 |
| Cd (µg g$^{-1}$) | 8.83 | 0.30 | 0.40 | 0.60 |
| Cr (µg g$^{-1}$) | 53.85 | 150 | 200 | 250 |

Note: '--' in the table indicates zero addition.

*2.2. Sampling and Measurement Methods*

2.2.1. Substrate Chemical Properties

Ten representative plants and the substrate samples from each treatment were collected 35 days after sowing and taken back to the laboratory (The Horticultural Laboratory at the College of Agriculture, Ningxia University, Yinchuan, China). The substrate samples were dried naturally (15 days) and then sieved through 1.0 mm or 0.5 mm sieves to obtain dry substrate samples of 1.0 mm or 0.5 mm diameter. A substrate fraction of 0.5 mm was used for the determination of total N, total P, total K, organic matter, and heavy metals. A substrate fraction of 1 mm was used to measure the available N, available P, available K, EC, and pH. One mm substrate was used, and the pH and EC were measured using a

pH meter (FE28, Mettler Toledo, Shanghai, China) and an EC meter (FE30, Mettler Toledo, Shanghai, China), respectively. The substrate and water were combined at a ratio of 1:10 and were shaken for one hour using an orbital shaker (HY-5A, Changzhou Guowang Instrument Manufacturing Co., Ltd. (Changzhou, China)). The organic matter (OM) content of the substrate sample was determined using the potassium dichromate sulfite oxidation method [25]. The total N content was assayed using the $H_2SO_4$-$H_2O_2$ digestion semi-micro Kjeldahl method [26], total P content was analyzed using the $H_2SO_4$-$H_2O_2$ digestion molybdenum antimony colorimetric method, and total K content was determined using the $H_2SO_4$-$H_2O_2$ digestion flame photometric method [27]. The available N content was determined using the alkaline diffusion method, and the available P content was determined using the $NaHCO_3$ leaching-molybdenum blue colorimetric method [28]. The available K was determined using ammonium acetate leaching and flame photometry [29].

### 2.2.2. Measurement of Plant Growth Indicators

Five days after sowing, the seedling emergence rate was counted. At 35 days after sowing, 10 representative plants were randomly selected from each treatment to determine plant height, stem diameter, and chlorophyll content. Plant height was measured using a tape measure from the shoot apical meristem to the root/hypocotyl junction, stem diameter was measured using a vernier caliper 1 cm below the cotyledons, and chlorophyll was measured using a chlorophyll meter (SPAD-502 plus, Konica Minolta, Tokyo, Japan) on the fourth leave of the plant (from cotyledons to shoot apical meristem). The representative plants were then harvested. Aboveground (from the root/hypocotyl junction to the shoot apical meristem) and belowground parts (from the root/hypocotyl junction to the root apical meristem) were separated and washed with deionized water, blotted dry with absorbent paper, and weighed using a single-pan analytical balance (ESJ200-5A, Shanghai Precision Instruments Co., Ltd., Shanghai, China) to determine the fresh weight. The plants were then placed in an oven (ZF-9210, Changzhou Guowang Instrument Manufacturing Co., Ltd.; Changzhou, China) at 105 °C for 15 min and then dried at a temperature of 80 °C (24 h) to a constant weight.

### 2.2.3. Determination of Heavy Metal Content

The dried plants were ground and sieved through a 0.5-mm sieve. The dried samples were mixed with strong acid, and the heavy metals were extracted using a microwave extractor (Master-40, Shanghai Xinyi Microwave Chemistry Technology Co., Ltd.; Shanghai, China). The heavy metal content in the substrate and plants was determined using inductively coupled plasma emission spectrometry (ICP-OES). Arsenic (As), manganese (Mn), nickel (Ni), copper (Cu), zinc (Zn), chromium (Cr), lead (Pb), and cadmium (Cd)⁻ and the heavy metal content of substrate W was determined as follows [30]:

$$W = C \times V/M \quad (1)$$

where $C$ represents the content of a heavy metal measured using an inductively coupled plasma spectrometer (mg/L); V represents the volume of the test solution (mL); and M represents the weight of the samples (g).

### 2.2.4. Calculation of the Enrichment and Transfer Coefficient of the Heavy Metals

The suitability of plant growth in heavy metal environments was evaluated by calculating the enrichment and transfer coefficients of heavy metals in the plants. The enrichment coefficient is an index for plant enrichment of heavy metals, indicating the ability of the plant to absorb heavy metals from the soil environment. The transfer coefficient indicates the ability to transfer a heavy metal from the belowground to the aboveground parts of the plant. A higher transfer coefficient indicates that the heavy metal is more easily transferred from the belowground to the aboveground of the plant. The enrichment and transfer coefficients were calculated as follows [30]:

$$BCF = C_{plant}/C_{soil} \qquad (2)$$

where BCF indicates the enrichment coefficient, $C_{plant}$ represents the amount of heavy metal in the belowground parts of a plant, and $C_{soil}$ expresses the amount of heavy metal in the soil.

$$BTF = C_{aboveground}/C_{belowground} \qquad (3)$$

where BTF indicates the transfer coefficient, and $C_{aboveground}$ and $C_{belowground}$ represent the heavy metal content in the aboveground and belowground parts of a plant, respectively.

### 2.3. Statistical Analysis

All experiments were replicated three times, and statistical analyses were carried out using Excel 2020 (Microsoft Corp., Redmond, WA, USA) and SPSS 25.0 (IBM Corp., Armonk, NY, USA). A one-way ANOVA was used for the tests, and significance was set at $p < 0.05$. LSD was used for the variance homogeneity test with Duncan's method for the significance analysis of multiple differences. Principal component analysis was used to analyze the relationships between substrate nutrient compounds. Membership functions were used for the comprehensive evaluation of substrate chemical properties. The figures were drawn using Origin 2018 (OriginLab; Northampton, MA, USA) software.

## 3. Results

### 3.1. Cucumber Seedling Growth

The seedling emergence rates in all treatments reached greater than 70%, and GFC0-GFC4 were higher than CK (Table 3). The highest rate of 93% was found in the GFC4 treatment, while the lowest rate of 74% was found in the GFC8 treatment. Compared to CK, the aboveground dry weight, the value of soil and plant analyzer development (SPAD), and stem diameter of the cucumber seedlings in the GFC0 treatment did not differ, while the belowground dry weight and plant height showed a decrease (Table 3). Adding sodium silicate to the seedling substrate containing GFC resulted in an overall decrease in the belowground dry weight and plant height of the cucumber seedlings. The GFC2 and GFC4 treatments significantly increased the SPAD values by 12.36% and 13.85%, respectively, when compared with the GFC0 treatment.

**Table 3.** Cucumber seedling growth indices in different treatments.

| Treatments | Emergence Rate | Overground Dry Weight (g) | Underground Dry Weight (g) | Chlorophy ll | Plant Height (mm) | Stem Diameter (mm) |
|---|---|---|---|---|---|---|
| CK | 0.86 ± 0.04 ab | 0.28 ± 0.01 a | 0.03 ± 0.01 a | 35.52 ± 1.18 b | 84.58 ± 4.18 a | 2.85 ± 0.11 a |
| GFC0 | 0.89 ± 0.10 a | 0.28 ± 0.08 a | 0.03 ± 0.00 b | 38.04 ± 1.18 b | 74.18 ± 1.69 b | 3.13 ± 0.07 a |
| GFC2 | 0.87 ± 0.07 a | 0.22 ± 0.05 a | 0.02 ± 0.00 c | 42.74 ± 2.43 a | 65.73 ± 4.20 c | 2.95 ± 0.10 a |
| GFC4 | 0.93 ± 0.06 a | 0.21 ± 0.02 a | 0.02 ± 0.00 c | 44.31 ± 1.99 a | 46.92 ± 1.56 d | 3.02 ± 0.14 a |
| GFC8 | 0.74 ± 0.03 b | 0.20 ± 0.03 a | 0.02 ± 0.00 c | 38.93 ± 2.13 b | 49.25 ± 1.20 d | 2.97 ± 0.26 a |

Note: Different lowercase letters indicate significant differences between different treatments ($p < 0.05$).

### 3.2. Changes in the Substrate Nutrient Content

The N uptake amount played a decisive role in the accumulation of dry matter in the plants. Compared with CK, the GFC2 treatment reduced the total N content of the substrate by 45.64%, while the other three treatments showed no significant difference (Figure 1A). The organic matter content of the substrate increased proportionally to the amount of sodium silicate. Depending on the volume of sodium silicate that was added, i.e., 2 g/kg$^{-1}$ (GFC2), 4 g/kg$^{-1}$ (GFC4), and 8 g/kg$^{-1}$ (GFC8), the OM content increased by 11.91%, 26.79%, and 53.58%, respectively, when compared with the GFC0 treatment (Figure 1B). Adding GFC alone had no significant effect on the organic matter content of the substrate. Potassium plays an important role in regulating the osmotic potential of plant cells and also plays a role in activating the enzymes used during photosynthesis [31]. The addition of GFC and sodium silicate reduced the total K content of the substrate compared

with CK. Excluding the GFC4 treatment, the total K content decreased with increasing amount of sodium silicate (Figure 1C). The P content in the substrate indirectly affects the chlorophyll content, which directly affects the normal growth and development of plants [32]. With the exception of the GFC0 treatment, all treatments significantly increased the total P content of the substrate when compared with CK (Figure 1D). The total P content in the substrate increased significantly with GFC0 and GFC2 by 53.34% but decreased with GFC4 and GFC8 (Figure 1D, Table S3).

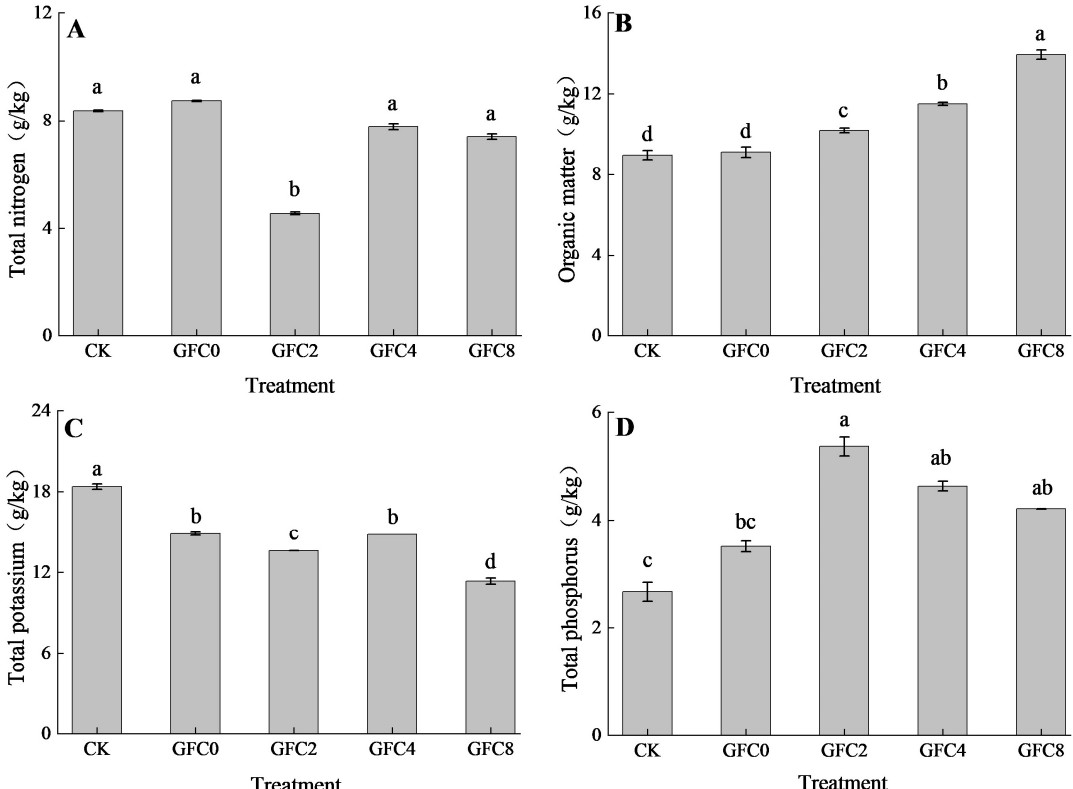

**Figure 1.** Effects of various matrix treatments on the total nutrient and organic matter content in the matrix. (**A**) Total nitrogen content (g/kg). (**B**) Organic matter (g/kg). (**C**) Total potassium (g/kg). (**D**) Total phosphorus (g/kg). Abbreviations: CK: compound matrix; GFC0: compound matrix + gasification filter cake; GFC2: compound matrix + gasification filter cake + 2 g/kg sodium silicate; GFC4: compound matrix + gasification filter cake + 4 g/kg sodium silicate; GFC8: compound matrix + gasification filter cake + 8 g/kg sodium silicate. Different lowercase letters indicate significant differences between different treatments ($p < 0.05$). LSD and Waller–Duncan were used for the variance homogeneity test and significance analysis, respectively.

The available N content in the substrate was measured and found to be increased by 20.82% in the GFC4 treatment but reduced by 18.80%, 10.27%, and 36.84% in the GFC0, GFC2, and GFC8 treatments, respectively, when compared with CK (Figure 2A, Table S4). The substrate available P content was highest in the GFC0 treatment (16.58%) and was reduced by 14.27% in the GFC4 treatment (Figure 2B). There was no significant difference between the other treatments when compared to CK (Figure 2B). The available K content in the substrate was higher in all treatments when compared with CK, where the GFC8 treatment was the highest (60.92%), followed by the GFC0, GFC2, and GFC4 treatments (Figure 2C). There were no significant differences between the GFC2 and GFC4 treatments (Figure 2C). The addition of both GFC and sodium silicate increased the pH of the substrate (Figure 2D). Adding GFC increased the EC in the substrate up to GFC2, but it decreased with the addition of sodium silicate (Figure 2D).

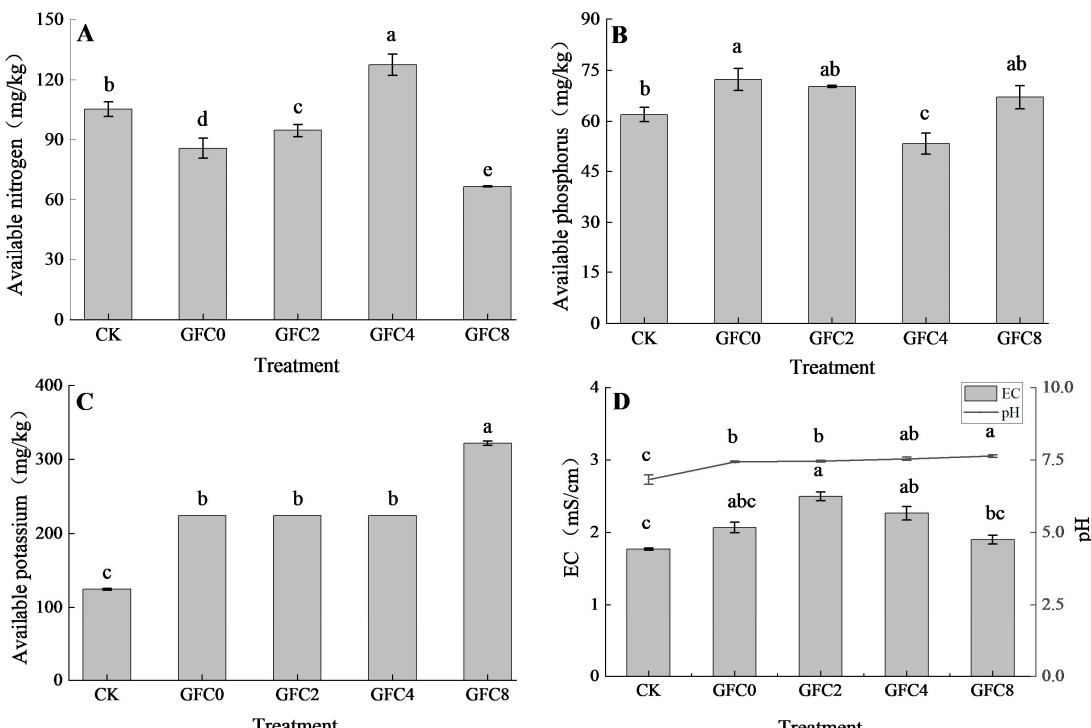

**Figure 2.** Effects of various matrix treatments on the available nutrients, matrix pH, and EC content in the matrix. (**A**) Available nitrogen (mg/kg). (**B**) Available phosphorus (mg/kg). (**C**) Available potassium (mg/kg). (**D**) EC (mS/cm). Abbreviations: CK: compound matrix; GFC0: compound matrix + gasification filter cake; GFC2: compound matrix + gasification filter cake + 2 g/kg sodium silicate; GFC4: compound matrix + gasification filter cake + 4 g/kg sodium silicate; GFC8: compound matrix + gasification filter cake + 8 g/kg sodium silicate. Different lowercase letters indicate significant differences between different treatments ($p < 0.05$). LSD and Waller–Duncan were used for the variance homogeneity test and significance analysis, respectively.

### 3.3. Heavy Metal Content in the Substrate and Plants

During the determination of heavy metals present in the substrate and plants, only Cr, Pb, and Cd were detected (Figure 3). The levels of As, Mn, Ni, Cu, and Zn were not measured as they were lower than the minimum detection limit (minimum detectable mass 0.5 ng). The content of Cr was higher than that of Pb in all substrates, while the content of Cd was the lowest (with the exception of Pb content in GFC2 being slightly higher than that of Cr) (Figure 3).

According to the requirements of the National Standard 15618-1995 [33] (Table 2), secondary-level soils (suitable for agricultural cultivation) must meet the pH values of 6.5–7.5 and a Pb, Cd, and Cr content less than 300 mg/kg, 0.4 mg/kg, and 200 mg/kg, respectively. The substrate in all treatments did not exceed the standard for Pb and Cr; however, the Cd contents exceeded the standard. All heavy metal contents exceeded the standard by above 90% (Table 2).

To evaluate the heavy metal content in cucumber seedlings, the GFC0–GFC8 treatments were analyzed. It was found that the GFC2, GFC4, and GFC8 treatments had higher levels of Cr, Pb, and Cd in the substrate compared with the GFC0 treatment (Figure 3C, Table S1). The substrate in the GFC2, GFC4, and GFC8 treatments had increased Cr contents by 12.10%, 30.83%, and 44.46%, respectively; increased Pb contents by 178.25%, 135.30%, and 115.63%, respectively; and increased Cd contents by 13.45%, 3.99%, and 10.83%, respectively, compared with the GFC0 treatment (Table S6). Compared with CK, GFC0 had a significantly increased amount of Cr, Pb, and Cd in the substrate by 40.61%, 73.34%, and 19.45%, respectively. The addition of sodium silicate to the substrate with added GFC significantly increased the levels of Cr, Pb, and Cd in the substrate. There was a trend indicating

increasing substrate Cr content with increasing amounts of sodium silicate, with significant differences detected between treatments. The Pb content in the substrate increased first and then decreased with increasing sodium silicate, with the highest Pb content in the GFC2 treatment, and significant differences were detected between treatments. All treatments significantly increased Cd content in the substrate compared with CK, but there was no significant difference among the treatments with added GFC (Figure 3C).

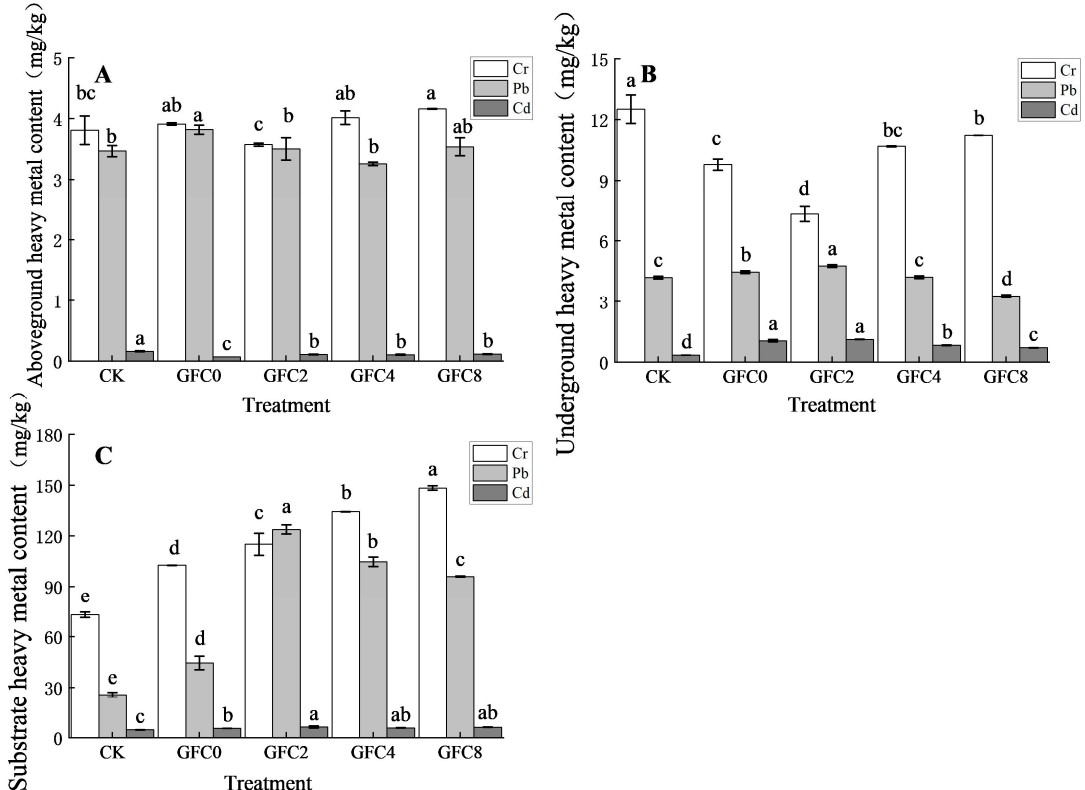

**Figure 3.** Contents of heavy metals in different parts of cucumber seedlings and in substrates. (**A**) Aboveground heavy metal content (mg/kg). (**B**) Belowground heavy metal content (mg/kg). (**C**) Substrate heavy metal content (mg/kg). Abbreviations: CK: compound matrix; GFC0: compound matrix + gasification filter cake; GFC2: compound matrix + gasification filter cake + 2 g/kg sodium silicate; GFC4: compound matrix + gasification filter cake + 4 g/kg sodium silicate; GFC8: compound matrix + gasification filter cake + 8 g/kg sodium silicate. Different lowercase letters indicate significant differences between different treatments ($p < 0.05$). LSD and Waller–Duncan were used for the variance homogeneity test and significance analysis, respectively.

The trend in heavy metal contents in the cucumber seedlings both aboveground and belowground was Cr > Pb > Cd for all GFC treatments (Figure 3A,B). Comparatively, the heavy metal content in the belowground parts was higher than that in the aboveground parts across all treatments (Figure 3A–C, Table S5). Specifically, the Cr content in the belowground parts was 2.13–3.27 times greater than that in the aboveground parts. The GFC8 treatment resulted in a Pb content that was 1.08 times higher in the aboveground parts than in the belowground parts. However, the GFC0 to GFC4 treatments had 1.17–1.40 times more Pb in the belowground parts compared with the aboveground parts, while Cd content in the belowground parts was 2.07–14.66 times higher than that in the aboveground parts (Figure 3A,B). Compared with CK, the GFC0 treatment increased the Pb content in the aboveground parts by 10.12% and increased the Cd content in the aboveground parts by 150.67%. Regarding the aboveground parts, the addition of sodium silicate increased the Cd content, the GFC2 and GFC4 treatments reduced the Pb content, and the GFC2 treatment reduced the Cr content when compared with GFC0. Regarding the belowground parts,

compared with CK, GFC2 reduced the Cr content, GFC8 reduced the Pb content, and GFC4 and GFC8 increased Cd (Figure 3A,B).

### 3.4. Enrichment and Transfer Coefficient of Heavy Metals

When analyzing the enrichment coefficients of the heavy metals, the Cr and Cd levels were higher (1.96–3.30 times and 2.07–14.66 times, respectively) in the belowground parts compared with the aboveground parts (Figure 3A,B). The enrichment coefficients for Pb were not significantly different between the aboveground and belowground parts (Figure 4C, Table S2). Compared with CK, all GFC treatments significantly reduced the enrichment coefficient of Cd in the aboveground parts of the plant but increased the enrichment coefficient of Cr in the belowground parts (Figure 4B). Thus, the enrichment coefficient for heavy metals, particularly Cr, decreased with increasing sodium silicate, while no significant difference was observed between GFC0 and GFC2 (Figure 4A–C).

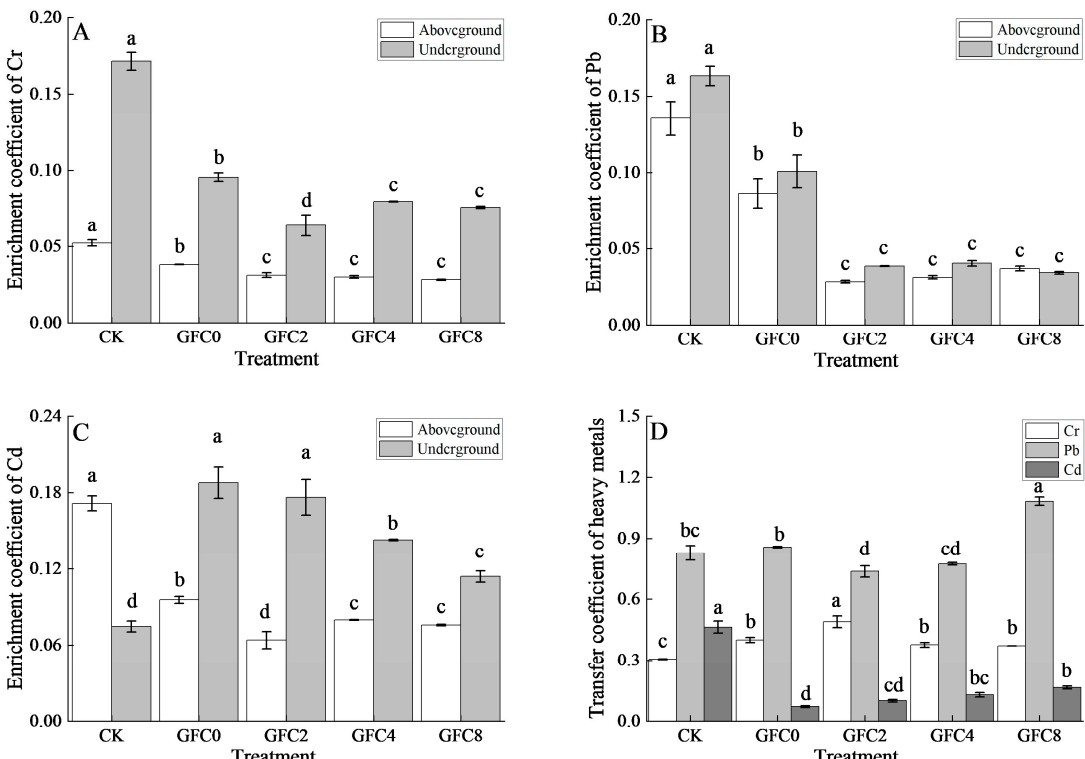

**Figure 4.** Transfer coefficients of heavy metals. (**A**) Enrichment coefficient of Cr. (**B**) Enrichment coefficient of Pb. (**C**) Enrichment coefficient of Cd. (**D**) Abbreviations: CK: compound matrix; GFC0: compound matrix + gasification filter cake; GFC2: compound matrix + gasification filter cake + 2 g/kg sodium silicate; GFC4: compound matrix + gasification filter cake + 4 g/kg sodium silicate; GFC8: compound matrix + gasification filter cake + 8 g/kg sodium silicate. Different lowercase letters indicate significant differences between different treatments ($p < 0.05$). LSD and Waller–Duncan were used for the variance homogeneity test and significance analysis, respectively.

Compared with the GFC0 treatment, the transfer coefficient of Cr increased with GFC0, and there was a significant increase of 22.50% for GFC2 (Figure 4D). This was followed by a decrease with the addition of sodium silicate, where there was no significant difference for the GFC4 and GFC8 treatments (Figure 4D). Compared with CK, the transfer coefficients of Cd increased by 40.85%, 83.10%, and 128.57%, respectively (Figure 4D).

### 3.5. PCA Analysis

To assess the relationship among chemical properties, heavy metal contents, and heavy metal enrichment and transfer coefficients in the substrate, a principal component analysis (PCA) was carried out (Figure 5). Principal component (PC) 1 and PC 2 were

extracted and found to explain 75.7% of the variation, with PC1 explaining 56.4% and PC2 explaining 19.3% of the variation. TN and TK contributed negatively to PC1, while OM, TP, pH, and EC contributed positively to PC1 (Figure 5A). In PC2, Cr, AP, and AK were negative contributors, while Pb, Cd, and AN were positive contributors (Figure 5A). The correlations among chemical properties, nutrition content, and heavy metal content are shown in Figure 5A. Organic matter was positively correlated with pH and AK, which indicated that adding sodium silicate led to an increase in pH in the substrate but not a loss of K. In the PCA, there was a clear separation among all treatments (Figure 5A). CK and GFC0 were in quadrant III, where there were higher levels of TK and TN content (Figure 5A). GFC2, GFC4, and GFC8 were in quadrants II and IV, where there were higher TP and heavy metal content (Figure 5A). Based on the PCA analysis, the comprehensive scores across the treatments showed that GFC4 was the highest followed by GFC2, GFC8, GFC0, and CK (Figure 5B).

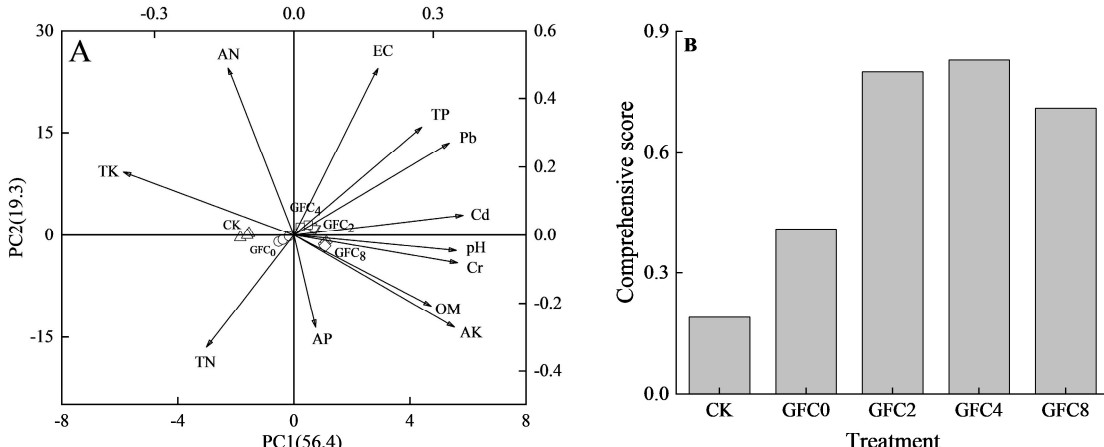

**Figure 5.** Principal component analysis of the chemical properties of the matrix and heavy metals. (**A**) PC1 and PC2. (**B**) The comprehensive score of each treatment. Abbreviations: AK, available potassium; TK, total potassium; TN, total nitrogen; AP, available phosphorus; AN, available nitrogen; OM, organic matter; TP, total phosphorus; CK: compound matrix; GFC0: compound matrix + gasification filter cake; GFC2: compound matrix + gasification filter cake + 2 g/kg sodium silicate; GFC4: compound matrix + gasification filter cake + 4 g/kg sodium silicate; GFC8: compound matrix + gasification filter cake + 8 g/kg sodium silicate. LSD and Waller–Duncan were used for the variance homogeneity test and significance analysis, respectively.

## 4. Discussion

In this study, we analyzed the effect of GFCs on cucumber seedlings. Previous studies have shown that GFC can be used as an adsorbent to reduce $NH_3$ volatilization during composting and can also improve the physical and chemical properties of the substrate, increase the nutrient content of the substrate, and improve the seed germination index [11,34]. However, since the inorganic components in GFCs cannot be melted, the carbon components cannot be fully burned. Therefore, a GFC has a huge specific surface area and contains many incompletely burned carbon particles, which has a similar effect as biochar [35–37].

This study showed that the addition of a GFC reduced the dry weight of the roots and decreased the plant height of the cucumber seedlings (Table 3). This is because a GFC has a higher content of total N and total K and a lower content of available N, compared with commercial substrate (Table 2). Furthermore, available N is positively correlated with plant growth. Soil available nitrogen content increases and then decreases with increasing pH. The GFC8 treatment, due to high pH, reduced the available nitrogen content in the substrate. In contrast, the substrate pH in the GFC4 treatment was more friendly to the available nitrogen [38,39]. Phosphorus can promote the formation and growth of early roots, thereby improving the ability of plants to adapt to external environmental conditions [40,41]. With increased sodium silicate, the available P content in the substrate

decreased first and then increased, with a minimum value detected in the GFC4 treatment (sodium silicate: substrate, at mass rates of 4 g/kg) (Figure 2). Interestingly, the result was opposite to the change in chlorophyll in the cucumber seedlings. The chlorophyll content in the cucumber seedlings increased first and then decreased with increasing addition of sodium silicate in the seedling substrate mixed with the GFC, and the maximum chlorophyll value was found in the GFC4 treatment (Table 3). This is likely because P is essential for chlorophyll synthesis, and the available P content in the substrate is depleted in the process of increasing the chlorophyll content of the plant [42,43]. Therefore, the amount of available P in the substrate is a key factor influencing the chlorophyll content in cucumber seedlings. Moreover, it may be that more P is used in plant absorption to meet crop growth. In addition, the content of heavy metals in the substrate also affects the chlorophyll content in plants [44]. Cadmium is easily absorbed and used by plants and can reduce the chlorophyll content in plants [45]. However, there was no significant difference in aboveground Cd content among the treatments (Figure 3). Therefore, Cd is not the key factor affecting the chlorophyll content in cucumber seedlings.

In this experiment, adding sodium silicate to the substrate enriched the heavy metal content in the substrate (Cr, Pb, and Cd) and reduced heavy metal transfer (Cr, Pb, and Cd) to the aboveground parts of the plant (Figures 3 and 4). Sodium silicate could reduce the enrichment of heavy metals in the shoots of the plants, yet different amounts of sodium silicate had different effects on the enrichment coefficients of the heavy metals. We found that with the addition of sodium silicate, the enrichment coefficients of Cr, Pb, and Cd were all higher in the belowground parts than in the aboveground parts of the plant (Figure 4A–C). The PCA revealed that the transfer of Cr, Pb, and Cd in the substrate was positively correlated with the pH of the substrate (Figure 5). This is because the substrate pH increased after the addition of sodium silicate, and a high soil pH value can reduce the effectiveness and migration of heavy metals in the soil [46,47]. Furthermore, a high soil pH promotes redox, adsorption, and organic complexation, which enhances the hydroxide precipitation of heavy metals [48,49]. The application of passivation agents (sepiolite, calcium–magnesium–phosphate fertilizer, and biochar)) increases soil pH and significantly reduces the effective state of soil Cd [50]. In a Cr stress study, the addition of exogenous silicon increased rhizosphere soil pH and accelerated the combination and fixation of active Cr with organic matter and carbonate, thus leading to reduced uptake of Cr by the roots [18,51].

The results from this study revealed that the content of heavy metals was higher in the belowground parts than in the aboveground parts (Figure 3A,B). This is due to the enrichment of heavy metals, primarily in the lower part of the plant. With an increasing amount of sodium silicate in the substrate, the Cr content decreased first and then increased in both the aboveground and belowground parts (Figure 3). This is due to the negative effect of sodium silicate, which causes the pH of the substrate to rise [52]. A high pH environment disrupts the cell membrane structure of the root system and allows the plant to take up higher levels of Cr, thus leading to a negative effect of sodium silicate [53]. At the same time, the aboveground Pb content in the cucumber seedlings decreased first and then increased, while the belowground Pb content increased first and then decreased (Figure 3). This is because low concentrations of Pb are rapidly precipitated as Pb-phosphate when taken up by protoplasts in the roots [54,55], reducing the amount of Pb transferred to the aboveground parts and reducing the damage caused by Pb to the aboveground parts [51]. However, the Pb content and Pb transfer coefficient exhibited an increasing trend in the aboveground parts of the cucumber seedlings after the addition of too much sodium silicate (Figures 3 and 4). This may be due to the high amount of sodium silicate that increased the pH of the substrate, which may have damaged the plant root cell membranes [56]. The refractory Pb-phosphate becomes microsoluble, resulting in increased Pb absorption by the cucumber seedlings. Furthermore, the addition of sodium silicate to the GFC substrate increased the aboveground Cd content in the cucumber seedlings, as Cd is readily absorbed by the plant. However, the Cd content increased first and then decreased in the

belowground parts (Figures 3 and 4), indicating that a suitable amount of sodium silicate mitigated the damage caused by Cd through the absorption and accumulation of Cd in the plant roots.

A mass rate of 4 g/kg of sodium silicate had the best effect on promoting plant growth and alleviating heavy metal stress. First, the GFC0 treatment was higher than CK, indicating that the GFC had some improvement effect on the substrate quality. Second, the comprehensive score in the GFC4 treatment was higher than in the GFC2 treatment, and both treatments scored higher than the other treatments. This indicated that the addition of a suitable amount of sodium silicate could improve the substrate quality, while an excessive addition of sodium silicate could have a negative effect on the substrate quality. An appropriate substrate pH value (from 6.0 to 7.5) is necessary for vegetable growth [57,58]. The addition of 8 g/kg sodium silicate increased the substrate pH value. This is because sodium silicate is a strong alkaline and weak acid salt ($11 < pH < 13$).

## 5. Conclusions

The addition of a GFC had no significant effect on the aboveground dry weight, chlorophyll, stem diameter, or seedling emergence of the cucumber seedlings or on the total N and organic matter content of the substrate. Adding appropriate amounts of sodium silicate significantly increased the SPAD value of the cucumber seedlings and the Pb and Cd contents in the substrate but reduced the Cr and Pb content in the aboveground and belowground plant parts. In addition, an appropriate amount of sodium silicate effectively fixed the heavy metal content in the substrate, inhibited the uptake of heavy metals by the plants, and alleviated the heavy metal stress. The addition of sodium silicate effectively relieved the risk posed by heavy metals in the GFC. Sodium silicate at 4 g/kg had the best effect on promoting plant growth and alleviating heavy metal stress followed by 2 g/kg sodium silicate. In summary, the composite application of sodium silicate and a GFC not only prevents heavy metal pollution but also does not affect the increase in nutrient content of the GFC to the substrate, which greatly improves the application potential of GFCs as a substrate.

**Supplementary Materials:** The following supporting information can be downloaded at: https://www.mdpi.com/article/10.3390/horticulturae9090988/s1, Table S1: The heavy metal enrichment coefficient; Table S2: The heavy metal transfer coefficient; Table S3: Effects of various matrix treatments on the total nutrient and organic matter content of the matrix; Table S4:Effects of various matrix treatments on the available nutrients, matrix pH, and EC of the content of the matrix; Table S5: Contents of heavy metals in different parts of the cucumber seedlings; Table S6: Contents of heavy metals in substrates.

**Author Contributions:** Conceptualization, W.T. and X.W. (Xiaodong Wang); methodology, Z.L. and K.G.; software, W.T. and Z.L.; validation, W.T. and K.G.; formal analysis, W.T. and X.W. (Xiaodong Wang); resources, X.Z.; original—draft preparation, W.T. and K.G.; writing—review and editing, X.Z., Z.L., and W.T.; visualization, W.T. and S.S.; supervision, X.Z. and X.W. (Xiaozhuo Wang); project administration, X.Z.; funding acquisition, X.Z. All authors have read and agreed to the published version of the manuscript.

**Funding:** This study was supported by the Key Research and Development Program of Ningxia (No. 2021BBF03002), the Science and Technique Innovation Leader Program of Ningxia (No. KJT2017001), and the Ningxia horticulture national first-class construction discipline project (No. NXYLXK2017B03). We are particularly grateful to the editor and anonymous reviewers for their help in improving our manuscript.

**Data Availability Statement:** Informed consent was obtained from all subjects involved in this study.

**Conflicts of Interest:** The authors declare that they have no known competing financial interest or personal relationships that could have appeared to influence the work reported in this paper.

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
