# Peer review of "Sodium Silicate Improves Cucumber Seedling Growth and Substrate Nutrients and Reduces Heavy Metal Accumulation in Plants"

_horticulturae, doi:10.3390/horticulturae9090988_

Round 1
Reviewer 1 Report
The research investigated the impact of sodium silicate on cucumber plant growth, substrate chemical properties and heavy metal accumulation. The topic is interesting and is very important, having the potential for a future publication. Some suggestions have been described below:
- Exclude the word “effect” from the Title.
- In the Abstract and Introduction, reorganize the objective of the work. Be more objective and short. As it is, it's confusing.
- Explore more the methodology in the Abstract. There is only one sentence citing the treatments. Detail this information further.
- Replace repeated keywords in the title.
- The last paragraph of Introduction needs to be rewritten. In this paragraph, only hypotheses and objectives should be written. This information, which is currently available, is part of the methodology. To review.
- I did not understand how only the ANOVA was done and there are letters in the figures. Was any average test done in addition to ANOVA?
- In the Results, a figure with main components is presented, however, in the methodology this analysis is not described. To review.
- In the Methodology it is mentioned that the Pearson correlation was performed. Where are these results???
Reviewer 2 Report
I regret to recommend that the manuscript is not acceptable for publication in its present form. All detailed comments and suggestions can be found in the attached file.

Minor editing of English language required.
Reviewer 3 Report
The effect of sodium silicate on cucumber seedling growth, substrate nutrients, and the accumulation and transfer of heavy metals in plants
Manuscript Number: horticulturae-2537401
In my opinion, the subject matter dealt with by the authors is very interesting. However, having thoroughly reviewed the manuscript presented to me, I have some minor comments and suggestions, which I present below:
1. The title of the manuscript is confusing. The author observed the effect of sodium silicate, GFC and caragana compost on the translocation of heavy metals and seedling growth of cucumber. Why does the title only focus on sodium silicate?
2. The composition of GFC and caragana compost is missing.
3. How and on what basis the dose of GFC and caragana compost was determined?
4. Which design was followed for setting up the experiment?
5. The extraction of heavy metals from plants and substrate is missing.
6. The GFC and compost might influence the adsorption and translocation of heavy metals to plants. Why only sodium silicate was considered as factor and one way ANOVA was performed? Why was compost not considered as factor and two-way ANOVA was not performed?
7. In line 186, the author referred Table 3, but I didn’t find any Table 2 and Table 3 in this manuscript.
8. In line 196, the authors are speaking about N uptake by plants but in the heading 3.2 they are speaking about nutrient composition of substate not the plants.
9. No proper explanation was given why the available N was significantly higher is GCF 4 and significantly lower in GCF 8 treatment?
Minor editing of English language required
Round 2
Reviewer 1 Report
The authors made the suggested modifications.
Author Response
Thank you very much for your recognition!
Reviewer 2 Report
For a clearer expression, I recommend you to reformulate the footer of Tables S1-S6 as follows: The values represent mean ± standard error. Different letters in the same column indicates significant differences (P < 0.05).
Minor editing of English language required.
Author Response
Dear Reviewer
Point 1: For a clearer expression, I recommend you to reformulate the footer of Tables S1-S6 as follows: The values represent mean ± standard error. Different letters in the same column indicates significant differences (P < 0.05).
Response 1: We thank you for pointing this out. We have reformulated the footer of Tables S1-S6 as your suggestion. Please see supplementary material.
Reviewer 3 Report
Thank you very to the authors for addressing all the comments and issues raised. I manuscript has been improved significantly to be consider for publication in reputed journal like Horticulturae.
Author Response
Thank you very much for your advice and recognition!